# Acupuncture Therapy for Military Veterans Suffering from Posttraumatic Stress Disorder and Related Symptoms: A Scoping Review of Clinical Studies

**DOI:** 10.3390/healthcare11222957

**Published:** 2023-11-14

**Authors:** Hui-Yong Kwak, Jungtae Leem, Hye-bin Seung, Chan-Young Kwon, Hye-Seon Jeong, Sang-Ho Kim

**Affiliations:** 1Republic of Korea Army, Capital Defense Command, Gwacheon-daero, Gwanak-gu, Seoul 08801, Republic of Korea; polestar92@daum.net; 2Research Center of Traditional Korean Medicine, College of Korean Medicine, Wonkwang University, 460 Iksan-daero, Iksan 54538, Republic of Korea; julcho@naver.com; 3College of Korean Medicine, Daegu Haany University, 1, Hanuidae-ro, Gyeongsan 38578, Republic of Korea; hyebin_73@naver.com; 4Department of Oriental Neuropsychiatry, College of Korean Medicine, Dong-Eui University, Busan 47227, Republic of Korea; beanalogue@naver.com; 5Department of Clinical Korean Medicine, Graduate School, Kyung Hee University, 26 Kyungheedae-ro, Dongdaemoon-gu, Seoul 02447, Republic of Korea; jhs0210@naver.com; 6Department of Neuropsychiatry of Korean Medicine, Pohang Korean Medicine Hospital Affiliated to Daegu Haany University, Pohang 37685, Republic of Korea

**Keywords:** armed conflicts, military personnel, posttraumatic stress disorder, acupuncture, scoping review

## Abstract

Military personnel in combat face a high risk of developing posttraumatic stress disorder (PTSD). In this study, a protocol-based scoping review was conducted to identify the current status of research on the efficacy of acupuncture for treating combat-related PTSD in military personnel. A literature search was conducted across 14 databases in November 2022, and data from the included studies were collected and descriptively analyzed. A total of eight studies were included. Participants were assessed for core PTSD symptoms using the PTSD Checklist for Diagnostic and Statistical Manual of Mental Disorders-5 and the Clinician-Administered PTSD Scale, as well as related symptoms, such as sleep issues. Although the efficacy of acupuncture has been substantiated in numerous studies, certain metrics did not exhibit improvement. Auricular acupuncture was the most commonly used treatment (50%) followed by manual acupuncture (25%) and a combination of both (25%). Shenmen and Kidney points were frequently targeted at auricular acupoints. The treatment period varied between 5 days and 2 months. While adverse events were reported in two of the fifty-five patients in the intervention group and in four of the sixty-four patients in the control group in the randomized controlled trial studies, no fatal adverse events were reported.

## 1. Introduction

Posttraumatic stress disorder (PTSD) is diagnosed when an individual who has in various ways been exposed to a severely threatening event displays a series of reactions and symptoms that persist for more than one month [1]. Military personnel who participate in combat are exposed to very threatening environments; thus, a high prevalence of PTSD is observed. Combat-exposed veterans have a notably high prevalence of PTSD, with a lifetime prevalence of 7.7% and 13.4% in males and females, respectively [2]. It is estimated that 10.1% [3] of Gulf War veterans and 15.8% [4] of Operation Enduring Freedom and Operation Iraqi Freedom veterans have PTSD.

Following deployment, veterans and active-duty military personnels with PTSD face numerous challenges, including reintegration into society and occupational difficulties. Studies have demonstrated that patients with PTSD have more significant challenges in regulating their emotions; moreover, a higher incidence of depression is observed in veterans with PTSD than those exposed to combat experiences without PTSD [5]. Furthermore, veterans with PTSD frequently report increased alcohol consumption, which can lead to alcohol use disorder [6]. Additionally, PTSD negatively affects familial relationships. Studies have revealed that as the severity of PTSD symptoms in veterans increases, parental functioning and marital adjustment decrease [7].

Currently, psychological treatments, such as prolonged exposure, cognitive processing, and cognitive behavioral therapies, are strongly recommended for treating PTSD [8]. Nevertheless, the stigma associated with mental illness among military personnel can hinder their willingness to seek psychological help [9], potentially creating a barrier to accessing mental health care [10]. Negative attitudes toward psychotherapy can also contribute to treatment dropout [11].

While psychotherapy, which is considered the gold standard for PTSD, receives the highest level of recommendations [12], selective serotonin reuptake inhibitors (SSRIs) are considered the most effective pharmacological treatment [13]. However, long-term administration of SSRIs can have numerous side effects, such as hyponatremia, genitourinary problems, hepatotoxicity, weight gain, sexual dysfunction, gastrointestinal problems, and central nervous system problems [14]. Withdrawal syndrome has also been reported after discontinuing the use of SSRIs [15]. 

Acupuncture, a non-pharmacological intervention that originates in the East and involves the insertion and manipulation of needles at the acupoint, has gained increasing attention. Most acupoints are effective as they belong to the meridian associated with each functional system of the human body [16], and practitioners take this into consideration when formulating acupuncture prescriptions. Recently, research has shown that acupuncture point stimulation has a significant effect on the central nervous system, including the limbic system and subcortical grey structures [17]. Preclinical animal studies utilizing acupuncture to treat animal models of PTSD have shown promising results, including improvements in anxiety, fear, sleep disturbances, cognitive symptoms, and depression [18]. Acupuncture is now classified as a non-pharmacological and non-psychological treatment option for PTSD [19]. A recent meta-analysis reported that acupuncture significantly improved PTSD and depressive symptoms, with effects lasting throughout the follow-up period [20]. 

Auricular acupuncture, in particular, has been widely disseminated through the use of specific manuals, such as the Battlefield Acupuncture (BFA) and National Acupuncture Detoxification Association (NADA) protocols. These protocols have been developed and applied in pain management [21] and addiction treatment [22]. BFA is specifically designed for the simple treatment of various types of pain in veterans [23] and active-duty military personnel [24], making it an efficient option on the battlefield. Auricular acupuncture, which has been used in various disaster and combat settings, has been shown to improve the symptoms of moderate PTSD [25].

Despite the considerable efficacy of acupuncture in the treatment of PTSD in military medicine, as described above, no systematic or scoping review of acupuncture studies on PTSD occurring during combat in military personnel has been published. A scoping review is a methodology used to identify certain characteristics or concepts of studies and conduct mapping and discussion about these characteristics or concepts, rather than presenting answers to specific questions, whereas a systematic review is structured to find answers to sophisticated clinical questions. A scoping review is often considered a preliminary step for identifying questions suitable for a systematic review [26]. In the present context, our understanding of the application of acupuncture in veterans with PTSD is limited. Instead of having a clear clinical question, we recognize the need to comprehend the current state of research by collecting and reconstructing overall knowledge. 

In this paper, we thoroughly examined the clinical research design and methodological characteristics of acupuncture, such as the characteristics of research participants, studied outcomes, and treatment regimens. In this study, we aimed to identify existing research gaps and the need for additional research and to extract appropriate clinical questions for a systematic review to be conducted in the future. 

## 2. Materials and Methods


**Study Design and Registration**


This study employed the scoping review process outlined by Arksey and O’Malley [27] and other relevant sources [28,29]. The scoping review protocol adhered to the Preferred Reporting Items for Systematic Reviews and Meta-Analysis (PRISMA) Extensions for Scoping Reviews (PRISMA-Scr) criteria. The review protocol was registered with the Open Science Framework (https://osf.io/t723f/, accessed on 13 June 2022) to ensure transparency on 13 June 2022, and updated on 2 November 2022. The protocol article [30], which described the research strategy for this scoping review, was released in January 2023 and provided a detailed explanation of the research techniques used in the study.


**Stage 1: Identifying the Study Questions**


The research team consisted of a clinical research specialist in traditional East Asian medicine (JL), an undergraduate researcher (HBS), and three neuropsychiatry specialists (SHK, CYK, and HYK). The team conducted a preliminary literature search to identify previous clinical evidence and worked collaboratively to develop and refine the research questions to ensure the thoroughness of the scoping review. The research team formulated the following questions to guide the scoping review:What are the characteristics of the studies that have investigated the use of acupuncture for managing PTSD in military veterans, including research design and target population?What clinical outcomes have been examined in previous studies on PTSD management in military veterans?What is the acupuncture therapy regimen for managing PTSD in military veterans?What is the effectiveness and safety of using acupuncture for treating PTSD in military veterans, as reported in previous studies?


**Stage 2: Identifying relevant studies**



**Information Source**


In stage 2, we identified relevant studies through a literature search of peer-reviewed articles on the use of acupuncture for PTSD. The search covered the period from the beginning of each database until November 2022. We used several databases, including MEDLINE (via PubMed), Embase, Cochrane Central Register of Controlled Trials, Allied and Complementary Medicine Database, Cumulative Index to Nursing and Allied Health Literature, Web of Science, Scopus, PsycArticles, Wanfang, VIP, China National Knowledge Infrastructure, Korea Citation Index, Oriental Medicine Advanced Searching Integrated System, and Citation Information by NII. Additionally, we considered grey literature searches, such as conference proceedings and doctoral dissertations, using Google Scholar. We manually searched the reference lists of relevant reviews and articles and contacted the authors of published papers with inaccessible electronic files. The search strategy was developed upon consultation with a clinical researcher, literature review expert, and psychiatric disease specialist. We used a combination of synonyms and related medical subject headings for the disease terms related to PTSD caused by war exposure, and intervention terms related to acupuncture. Detailed search terms and strategies are provided in Appendix A.


**Eligibility Criteria:**


(1) Study design: In this scoping review, we considered clinical research studies that investigated the effects of acupuncture intervention on military veterans with PTSD. Randomized controlled clinical trials, quasi-randomized controlled trials, non-randomized controlled trials, single-arm trials, case series, cross-sectional studies, and feasibility studies were also included. We excluded case reports with less than three patients [31], literature reviews, and preclinical studies. We examined the reference list of each selected systematic review. (2) Participant type: The participants eligible for this scoping review were military veterans who had been exposed to combat and had PTSD symptoms. In this paper, “veteran” is defined as a military personnel who has participated in war or combat, irrespective of the current active or retired status. The studies included in this review used standardized diagnostic criteria for PTSD, such as those found in the Diagnostic and Statistical Manual of Mental Disorders (DSM) or the International Classification of Diseases. Additionally, studies that used validated PTSD evaluation tools, such as the Clinician-Administered PTSD Scale, PTSD Checklist, and Impact of Event Scale-Revised, with cutoff values as inclusion criteria were also included. The review also considered studies that included PTSD comorbid with medical illnesses, such as tinnitus and traumatic brain injury. (3) Intervention type: To be eligible for inclusion in the review, various types of acupuncture therapy were considered as possible interventions, including manual acupuncture, warm-needle acupuncture, fire needle acupuncture, electroacupuncture, pharmacopuncture, bee venom acupuncture, and acupotomy. However, acupressure therapy was not included in this review. The review did not restrict concomitant treatment, except for East Asian traditional medicine interventions, such as herbal medicine, moxibustion, cupping, and tui-na. Any type of intervention was permitted in the control group. There were no restrictions on the treatment period, dosage, or frequency of acupuncture. (4) Types of Outcome Measurements: Our review considered various symptoms that may arise after a diagnosis of PTSD. We did not impose any restrictions on the outcome variables considered. Based on a previous study [26], the outcomes were categorized into the following groups: (1) psychological symptoms, such as anger, irritability, guilt, anxiety, fear, distrust, sadness, shame, apathy, alienation, frustration, loss of confidence, and mourning; (2) somatic symptoms, such as palpitations, pain, anorexia, insomnia, and fatigue; and (3) cognitive symptoms, such as repeated recall of traumatic events, decreased memory, difficulty in making decisions, and lack of concentration. Due to the nature of the scoping review, the search and selection were conducted in an expansive manner with the agreement of the researchers, to view the research field as broadly as possible. In addition to these outcome variables, we investigated the safety issues including the incidence of adverse events and dropout rates.


**Stage 3: Study selection**


The research team established and agreed upon the inclusion and exclusion criteria. Three independent reviewers (HBS, HYK, and JHS) screened the articles, starting with a review of the titles and abstracts, to identify potentially relevant articles. A full-text review of articles that met the inclusion criteria was conducted. All the reasons for inclusion or exclusion were recorded according to the predetermined criteria. Any discrepancies between the reviewers were resolved through discussion with an independent researcher as a mediator.


**Stage 4: Charting the data**


The study team developed a data extraction sheet using pilot testing and multiple iterations. This sheet included various items to be collected from the studies, such as: (1) general information about the study, including the name of the first author, publication year, country, and research design; (2) participants’ demographic data, including the sex, age, number of participants (initial and final), disease duration and severity, and diagnostic criteria; (3) details of the intervention, such as the type of acupuncture, treatment dosage, treatment period, acupuncture point, and control/concomitant intervention; and (4) outcome variables regarding the efficacy and safety.

Three reviewers (HBS, HYK, and JHS) separately extracted data from the articles and crosschecked their findings. In case of any discrepancies, a third researcher (SHK) provided justifications to resolve the disagreements. This process ensured accurate and reliable data extraction.


**Stage 5: Collating, summarizing, and reporting the results**


The data extracted from the included studies were used to collate, synthesize, and summarize information according to the analytical framework of the scoping review. The qualitative data analysis phase included information, such as the authors’ names, countries, publication years, number of participants, sex, age, research designs, and type of treatment or control group intervention. The results of each outcome domain in the clinical research, the study’s conclusion, and the number of adverse events were included in the first table. The tables were categorized based on the study design, such as randomized controlled trials (RCT), before-after studies, case series, and qualitative studies. The second table, “Detailed Information of Acupuncture Treatment”, provided specific details about the intervention, including the type of acupuncture, location of the acupuncture point, treatment frequency, treatment period, treatment number, and details of the control or concomitant intervention. A research map table was used to visualize the current research status and frequently used outcomes, assisting researchers and practitioners in identifying knowledge gaps in PTSD-acupuncture research. A risk of bias assessment was not conducted, as the primary objective of this scoping review was to provide an overview and map the existing evidence, rather than undertaking a critical appraisal [26].

## 3. Results

### 3.1. Study Selection

Through a comprehensive database search, we retrieved 5532 records. After removing duplicates, 5136 records were considered. Following a title and abstract screening, 5103 records were excluded. We then retrieved the full text of the remaining 33 records for a detailed assessment, and 25 records were subsequently excluded for various reasons as shown in Figure 1. The list of full-text retrieved articles is provided in Appendix A. In total, eight studies, including four RCTs [32,33,34,35], two before–after studies [36,37], one case series [38], and one qualitative study [39], were included in this review (Figure 1). 

### 3.2. General Characteristics of the Included Studies

#### 3.2.1. RCTs

(1) Population: In the reviewed studies, enlisted personnel were the most common participant demographics, and more males than females were present in all studies. Three studies [32,33,35] enrolled veterans, two of whom were Operation Enduring Freedom and Operation Iraqi Freedom veterans [32,33], and two studies [32,34] enrolled active-duty military personnel. In three [32,33,34] of the four studies, the participants were diagnosed with PTSD based on the DSM-IV or DSM-IV-text revision criteria (Table 1). The mean age and standard deviation of the participants in each group are presented in Table 1, with mean values ranging from 30 to 40. (2) Intervention: Two studies [32,33] performed auricular acupuncture and one study [35] performed manual acupuncture. In one case [34], the usual PTSD care was implemented as concurrent therapy. The duration of the intervention varied from three weeks to two months. (3) Control: Interventions performed in the control group included sham acupuncture, usual PTSD care, and waiting-list control. Most studies [27,28,29] adopted a 1:1 design; however, Prisco et al. [33] compared the effects of a three-group design: manual acupuncture, sham acupuncture, and waiting list control. Engel, et al. [34] utilized an intervention of “acupuncture combined with standard PTSD care” in their study. The study design involved administering the usual PTSD care, including psychotherapy and medication, as per the clinical practice guidelines for PTSD, to a control group. The purpose of the study was to assess the added effects of acupuncture; thus, no efforts were made to increase adherence to standard care modalities. (4) Outcome: In three studies [27,28,30], sleep-related outcomes were the most commonly measured indicators. Objectively, sleep was measured using actigraphy, consensus sleep diary, and mini-sleep diary (MSD) [32,33,35]. Sleep was subjectively assessed using the Pittsburgh Sleep Quality Index (PSQI) [32,35] and Insomnia Severity Index (ISI) [33]. Changes in the PTSD symptoms were evaluated using the Posttraumatic Stress Disorder Checklist for DSM-5 (PCL) [32,34] and the Clinician-Administered PTSD Scale (CAPS) [34]. Depressive mood, pain, and quality of life were assessed [34] using the numeric rating scale (NRS), Beck Depression Inventory (BDI), Short Form 36 Physical Component Summary (SF-36 PSC), and Short Form 36 Mental Component Summary (SF-36 MCS), respectively (Table 2).

#### 3.2.2. Before-After Studies

(1) Population: In the two studies [36,37] we reviewed, participants were selected based on their experiences of war-related trauma, including combat-related trauma and military sexual trauma, as classified in the International Classification of Diseases, Tenth Revision (ICD-10), regardless of their PTSD diagnostic status (Table 1). One study [36] reported the mean age and standard deviation as 32.96 ± 9.66, while another study [37] reported an age range of 29–48. (2) Intervention: Manual acupuncture was applied in both studies, and auricular acupuncture was used in combination with manual acupuncture in one study. The duration of the intervention in the study conducted by Eisenlohr et al. [32] was not specified. On the other hand, the study by Cronin [37] involved a treatment period of 5 days with a follow-up assessment conducted 2 weeks post-treatment. (3) Outcome: In the study by Eisenlohr et al. [36], the evaluation of PTSD-related symptoms was based on five subjective rating scales, and no normalization was performed. In contrast, Cronin, and Conboy [37] utilized the PSQI to measure sleep quality and the PTSD Checklist for DSM-5 (PCL-M) to assess PTSD-related symptoms (Table 3).

#### 3.2.3. Case Series Study

(1) Population: The study population included patients previously diagnosed with PTSD who presented with tinnitus symptoms (Table 1). The age range of the participants in this study was 41–59 years, which was older than that reported in other studies. (2) Intervention: The intervention involved the application of manual acupuncture for an unspecified duration. (3) Outcome: The outcome was measured by recording and reporting the subjective improvement in the tinnitus symptoms as perceived by the patients (Table 4).

#### 3.2.4. Qualitative Study

(1) Population: The study population consisted of military personnel diagnosed with PTSD according to DSM-IV. (2) Intervention: The intervention applied was auricular acupuncture, which was administered over a period of three weeks (Table 5). (3) Outcome: The study collected and recorded the subjective responses of the participants related to the improvement of sleep disorders and their favorable perception of the auricular acupuncture procedure (Table 5).

### 3.3. Details of the Acupuncture Methods

The acupuncture methods used in the included studies were classified into three categories (Table 6).

(1) Auricular Acupuncture: Four studies [32,33,37,39] used auricular acupuncture as the sole intervention. In all the studies, the Shenmen and Kidney points were commonly selected, whereas the Heart and Sympathetic points were frequently utilized. The intervention was performed according to a set protocol, without operator discretion. The treatment time and frequency varied, with acupuncture being performed for over 30 min per session, and the visit interval was set to be more frequent than twice a week. One study [33] reported that the needles were inserted perpendicularly, and reached the ear cartilage to a depth where they could stand on their own without using a guide tube. No manipulation was reported in the same study, whereas the other three studies did not report any induced responses.

(2) Manual Acupuncture: Two studies [3,36,38] included in this review performed manual acupuncture alone. In a study conducted by Eisenlohr et al. [36], rough guidelines were provided in the study design, with the procedure performed at the operator’s discretion. Four needles were reportedly used in a study by Arhin, et al. [38], which employed the Korea Four Needle Technique, whereas other studies did not report the number of needles used. (3) Auricular and Manual Acupuncture: In two studies [28,29] that combined the two methods, rough guidelines were provided for manual acupuncture, and the operator performed the procedure considering the characteristics of each patient.

The Shenmen point was commonly used as the auricular acupoint, which was prevalent in all the studies that applied auricular acupuncture. In the four studies that used manual acupuncture, PC6 was used three times, whereas SP6 was used twice, indicating frequent usage. The number of other acupuncture points varied. No manipulation was reported in one study, whereas no other studies reported any induced responses.

### 3.4. Reported Effectiveness and Safety of Acupuncture for PTSD

#### 3.4.1. Effectiveness in RCTs

In a study by King et al. [31], a significant improvement was observed only in the PSQI-sleep quality and daytime dysfunction in the auricular acupuncture group, whereas no improvement in the objective sleep indices was observed. A similar outcome was found in a study by Prisco et al. [32], in which the control group was assigned to a waiting list or sham acupuncture. Only the ISI-Total score significantly improved in the treatment group after one month of treatment. Conversely, Huang et al. [34] found a significant improvement in the PSQI global score and objective sleep measures assessed by actigraphy. Engel et al. [33] reported a significant improvement in PTSD symptoms as measured by the PCL, CAPS, NRS, BDI, and SF-36 in the test group, who received acupuncture as an adjunct to standard PTSD care. However, King et al. [31] found no significant improvement in the PCL-M and PHQ-9 scores in the group receiving acupuncture compared with the sham acupuncture group (Table 2).

#### 3.4.2. Effectiveness in the Before-After Study

All the before-after studies demonstrated improvements in PTSD-related symptoms and sleep quality after a series of acupuncture interventions. In a study conducted by Cronin and Conboy [36], both the PSQI and PCL-M were found to demonstrate significant improvements compared to the pre-treatment measurements. Eisenlohr et al. [35] reported subjective improvement in the symptoms, such as sleep disorders, restlessness, agitation, frightfulness, and aggression, among patients, without performing a statistical analysis (Table 3).

#### 3.4.3. Effectiveness in the Case Series

All symptoms of tinnitus accompanying PTSD reportedly improved following a series of acupuncture sessions, with patients expressing satisfaction with the acupuncture treatment [37] (Table 4).

#### 3.4.4. Effectiveness in the Qualitative Study

King, Moore, and Spence (2016) [38] aimed to explore the factors that contribute to improved sleep in veterans with PTSD-related insomnia undergoing acupuncture in a follow-up to their previous study [31]. Direct improvement responses included a faster ability to fall asleep, longer sleep duration, ability to fall asleep during auricular acupuncture, reduced frequency of nightmares, and improvement in daytime functioning. Indirect effects included reduced pain and increased relaxation levels (Table 5).

#### 3.4.5. Safety of Acupuncture

Adverse events were reported in three [31,32,33] of the four RCTs. Engel et al. [33] reported no adverse effects in either the acupuncture treatment group or the control group. In two other studies [31,32], falls and withdrawals due to discomfort from acupuncture were reported as side effects in the acupuncture group. Among the 119 participants in the RCTs, adverse events occurred in two of the fifty-five patients in the intervention group and four of the sixty-four patients in the control group. In one before-after study [35], two of twenty-seven participants reported side effects, including feelings of drunkenness and excessive sleep; however, these adverse events were alleviated after the exclusion of specific acupoints in subsequent interventions (Table 2, Table 3 and Table 4).

## 4. Discussion

This scoping review aimed to investigate the types, methods, measurement indices, and effectiveness of acupuncture in managing trauma in military veterans by collecting relevant studies. Although the number of selected studies was limited, the analysis of eight studies provided an overview of the current state of research in the field.


**Main findings and interpretation**



**Question 1: What are the characteristics of the studies on the use of acupuncture for PTSD management in military veterans?**


Four RCTs, two before-and-after studies, one case series, and one qualitative study were identified. RCTs, which provide the highest level of evidence in evidence-based medicine, were the most commonly conducted studies. This may be owing to the fact that acupuncture is not a new treatment and has already been researched in various fields, including trauma-related mental and physical symptoms [20]. Researchers are now investigating its effectiveness in comparison with other interventions in certain settings through RCTs, rather than merely comparing before-and-after effects of acupuncture. For example, Engel et al. [33] conducted a study to examine changes in the improvement of PTSD symptoms based on whether acupuncture was performed in addition to usual PTSD care. The study design was highly consistent with that of the current clinical setting. In addition, a supplementary qualitative study [38] was conducted to verify the efficacy of acupuncture and further substantiate the research findings, building upon prior RCTs [31].

Data on the rank, sex, age, and whether the participants were on active duty or not were collected as participant characteristics. In terms of rank, enlisted service members were the most prevalent, and in terms of sex, males were more represented than females. This sex ratio can be explained by the specificity of the military profession. However, officer rank and female sex were also represented in the study to a certain degree. In terms of age, the average age of the recruited study participants was 30 years in studies targeting active-duty military personnel within the range of military enlistment and discharge age. Meanwhile, a study [38] focusing on retired veterans who had been treated for PTSD long ago exhibited a relatively high age range of 41–59 years.


**Question 2: Which clinical outcomes were adopted in previous studies on PTSD management in military veterans?**


Studies have shown that acupuncture can improve the core and related symptoms of PTSD (50%) and sleep quality (67.5%). As a subjective indicator of sleep, the PSQI was used in three cases, in addition to the use of the ISI and MSD. Wrist actigraphy was used as an objective indicator in three studies. Problems related to sleep were classified separately as “nightmare symptoms in PTSD” and were considered important enough to be mentioned separately in many medical guidelines. This can be a major factor in lowering the work capacity of military personnel, who must always be prepared and vigilant to maintain military readiness.

To evaluate PTSD-related symptoms, most studies used the PCL or PCL-M, while one study used a combination of the PCL-M and CAPS-IV. Some studies use the BDI, NRS, and SF-36 to measure PTSD-related symptoms. As PTSD is a well-defined condition with established measurement indices, the PCL-M is a preferred method for measuring the outcomes. Given that the symptomatic manifestations of PTSD are characterized by diverse mental and physical symptoms, some studies have utilized alternative indicators to measure the effectiveness of interventions targeting this condition [29,32].


**Question 3: What is the regimen of acupuncture therapy for PTSD management in military veterans?**


Auricular acupuncture (50%), manual acupuncture (25%), or a combination of both (25%) were administered. Although a standardized protocol was commonly employed in acupuncture studies, some degree of operator discretion was occasionally allowed. Electroacupuncture and scalp electroacupuncture, which have been reported in other studies on participants with PTSD [35,36] without military issues, were not used. This is because standardized procedures using minimal equipment are required in battlefield situations, in which PTSD occurs in large numbers [40].

Auricular acupuncture was used in this study. To date, BFA and NADA are two existing auricular acupuncture protocols. In particular, BFA is widely applied in the US military and has been reported to have a significant effect on the treatment of pain [41]; it has been used for managing insomnia in service members with promising effects [39,40]. Patients with PTSD generally complain of new or worsening pain [42], with insomnia as a core symptom [43].

The NADA protocol was developed as an alternative non-pharmacological treatment for addiction; however, some papers have reported the application of the NADA protocol to PTSD treatment for patients exposed to massive traumatic events [44]. In this review, we included a study by Cronin et al. [36], which applied the NADA protocol and reported improvements in PTSD symptoms and sleep disorders. According to a recent study, significant parasympathetic nervous system activity was identified when heart rate variability (HRV) was measured after needling Shenmen and Zero points [45], indicating the role of auricular acupuncture in alleviating hypersensitivity-related symptoms accompanying PTSD associated with parasympathetic activity. Notably, Shenmen was also the most frequently selected acupoint in the studies analyzed in this review.

Acupuncture was typically administered for approximately 30 min per session. Although the duration varied, most studies recommended a minimum of three weeks of treatment. However, intensive intervention, such as receiving daily treatment, was effective even after just 5 days [36].


**Question 4: What have previous studies reported with respect to the effectiveness and safety of using acupuncture for treating PTSD in military veterans?**


With regard to PTSD-related symptoms, a significant decrease was observed in three of the four reported studies. The difference between the groups was statistically significant compared to the control group. However, in the studies by King et al. [31], no significant change was observed compared to the control group. A systematic review of the studies that applied acupuncture to PTSD in adults [20] also performed a meta-analysis on six of seven selected cases demonstrating significant overall effect size]. Studies conducted by Engel et al. [33] and King et al. [31] were also included in this analysis.

In the case of sleep-related indicators, no change was observed in the objective indicators in the study by King et al. [31]; however, improvements in sleep quality and daytime function were confirmed using the PSQI. A similar response was reported by Prisco et al. [32]. In the study by Huang et al. [34], a significant improvement in the global PSQI score was found in the mild traumatic brain injury group, which was divided by stratified randomization based on the presence or absence of PTSD, compared to the sham acupuncture group. These results are meaningful because the analysis showed that the presence of PTSD did not significantly affect the degree of improvement in insomnia.

Reports indicated that despite a lack of improvement in the total sleep time, the sleep quality and daytime activity capacity improved, in contrast to existing animal experiments [18]. In animal experiments, the manual acupuncture group showed an increase in total sleep time compared to the control group. However, as this study was conducted on humans, it is possible that improved sleep quality allowed the participants to reduce their sleep time and engage in other activities. It is also important to consider the potential ceiling effect of sleep time in treating mild sleep disturbances, as the baseline of the participants’ total sleep time was close to the normal values. The use of polysomnography to measure electroencephalography could provide a more accurate understanding of why total sleep time remained stable but sleep quality and daytime fatigue improved.

No serious adverse events directly related to acupuncture were reported, and some side effects resolved after stopping the intervention for specific acupoints. Generally, side effects of acupuncture can include infection, damage to the internal organs, such as pneumothorax or heart injury, injury to the central or peripheral nerves [46], local discomforts such as bruising and pain, and other reactions such as orthostatic problems, worsening of symptoms, headache, and fatigue [47]. Vasovagal syncope, suicide attempts, hypertensive crises, and asthma attacks have been reported as potentially serious adverse events [47]. The side effects of auricular acupuncture tend to focus on infection, local discomfort, and autonomic reactions, such as dizziness, as it does not cause damage to the internal organs or central nervous system [48]. In the studies reviewed in this report, needle acupoints were not located in the trunk; only those in the auricular acupuncture and upper/lower extremities were used; therefore, the risk of serious damage to the internal organs was likely eliminated. Therefore, the adverse effects of auricular acupuncture and acupuncture for PTSD military personnel in distal acupoints are not fatal, and the side effects are within the controllable range of the operator.


**Clinical significance of this review**


This study represents the first scoping review on the effects of acupuncture on war-related trauma among military personnel. Acupuncture, as a non-psychological and non-pharmacological intervention, may help overcome barriers to PTSD treatment, such as social stigma and limited access to mental health care.

The findings of this scoping review can be a useful reference when designing acupuncture protocols for the treatment of PTSD in military personnel. The development of standardized acupuncture protocols for the treatment of PTSD in military personnel represents a significant step in this field. As noted previously, the variability in the acupuncture protocols used in the studies included in this scoping review highlights the need for more consistency in treatment administration. The development of standardized protocols could also help increase the reproducibility of results in future studies and facilitate the comparison of outcomes across different studies.


**Limitations of this review**


This study had several limitations. First, the inclusion criteria were broad, resulting in the inclusion of all personnel who reported war-related trauma. This method maximizes the number of studies included in the scoping review but may result in a heterogeneous degree of suffering among the participants. A study by Eisenlohr, Römer, and Zimmermann [35] specifically targeted patients meeting Criterion A of the ICD-10 PTSD criteria and found that 19 of 27 patients were diagnosed with PTSD. The study by Cronin, and Conboy [36] focused on veterans with combat stress-induced insomnia, and using the PCL-M assessment tool, found that four of the five participants had symptoms exceeding the reference point for PTSD, with one participant not meeting the criteria for full-blown PTSD. Huang et al. [34] included patients with mild traumatic brain injury, with >60% of patients diagnosed with PTSD. The results of the regression analysis confirmed that PTSD did not have an interaction effect, which compelled us to include this review in our study. In the study by Arhin et al. [38], only past PTSD diagnosis history was confirmed, and no new PTSD diagnosis was made at the start of treatment. Second, studies may have been missed in the systematic search process, particularly those published in languages other than English, Chinese, Korean, or Japanese. Third, owing to the variation in the acupuncture protocols among the selected articles, accurate evaluation of specific acupuncture methods was not possible. However, this limitation can be addressed through future standardization efforts. Finally, a consultation step, which would have been the final step in the scoping review, was not performed owing to the small number of veterans in the Republic of Korea and ethical concerns that may arise in obtaining a list of contactable veterans.


**Suggestions for Future Research**


Although RCTs are the gold standard for evaluating the effectiveness of acupuncture in treating PTSD symptoms, prospective observational studies may provide valuable data if RCTs are not feasible. It is important to standardize acupuncture protocols as much as possible to ensure consistency across studies on acupuncture in military personnel with PTSD. Acupuncture treatment involves the application of multiple acupuncture points, which necessitates the establishment of a standardized protocol. In principle, the best method is to establish the most effective protocol through a pilot study, which compares the effectiveness of several candidate protocols using various acupuncture point combinations, manipulation methods, treatment frequencies, and periods. However, if there is a lack of time or budget for conducting basic research, the Delphi method, which involves gathering expert opinions through a structured iterative process [49], can be adopted. We suggest combining acupuncture points based on the meridian theory, with Shenmen, Zeropoint, and Sympathetic points being the most frequently used auricular acupuncture points. Improvement can be expected with acupuncture treatment administered at intervals of approximately twice a week for a total of approximately 10 treatments.

Outcome indicators, including core PTSD symptoms and sleep-related indicators, should be consistent with existing studies. The sleep-related outcomes of previous studies [31,32,34] were mostly self-reported measurements with the exception of actigraphy. Therefore, there is a need to evaluate the mechanisms of acupuncture using objective measurements. For example, measures such as salivary cortisol [50] and HRV parameters [51] can be applied to assess physiological changes, while functional magnetic resonance imaging [52] can be used to evaluate brain activation areas. A polysomnography test [53] should be conducted to investigate the paradoxical improvement in sleep, which has not been explained in previous studies.

Regarding participant characteristics, it is worth noting that the composition of military populations often results in a higher proportion of male participants in research studies, with some investigations even being exclusively conducted with male participants. Consequently, it is imperative to undertake measures to mitigate any gender bias in participant recruitment and employ subgroup analyses to determine potential gender interaction effects. Explicitly specifying the service status of military personnel participating in the study, whether active or retired, is essential owing to significant differences in their motivation for recovery and work environment. As the term “veteran” is ambiguous and can indicate either a retired service member or a military personnel with combat experience, it is crucial to clearly indicate the current service status of participants in the research conducted. The accumulation of a sufficient number of studies can promote categorizing them into subgroups and facilitate analyses.

In military medicine, cost-effectiveness is a pivotal area of interest in public health policies [54]. Particularly, in the event of a wartime scenario, where a substantial number of patients with PTSD may emerge, the significance of cost-effectiveness is amplified. Therefore, whether acupuncture can serve as a cost-effective intervention for combat-related PTSD should be investigated. If found to be a cost-effective solution, acupuncture could be adopted as an authorized response measure for PTSD care in military medicine in various countries. In military settings, manuals produced by higher-level organizations are disseminated to lower-level units on the battlefield. Therefore, if additional research confirms the effectiveness of acupuncture interventions for combat-related (PTSD, it may be possible to produce and distribute a manual that outlines the appropriate protocol for use in military practice. Furthermore, the distribution of chart forms that record PTSD symptoms before and after acupuncture treatment could facilitate retrospective data collection.

## 5. Conclusions

The available research on acupuncture as a treatment for trauma in military veterans has mainly been conducted using a RCT design, along with a few before and after studies, case series, and qualitative studies. Considering the various acupuncture approaches, auricular acupuncture has been the most commonly used approach, with the Shenmen and Kidney points being frequently targeted. The main outcome measures were scales related to core PTSD symptoms and sleep. The results of these studies suggested that acupuncture may be a potential intervention for military personnel experiencing combat-related PTSD, either alone or in combination with conventional therapies. However, further research is warranted to establish its efficacy as a stand-alone treatment for PTSD in military personnel. As acupuncture is already being utilized in the field, research should be directed to gather further evidence by compiling various data, through diverse study designs.

## Figures and Tables

**Figure 1 healthcare-11-02957-f001:**
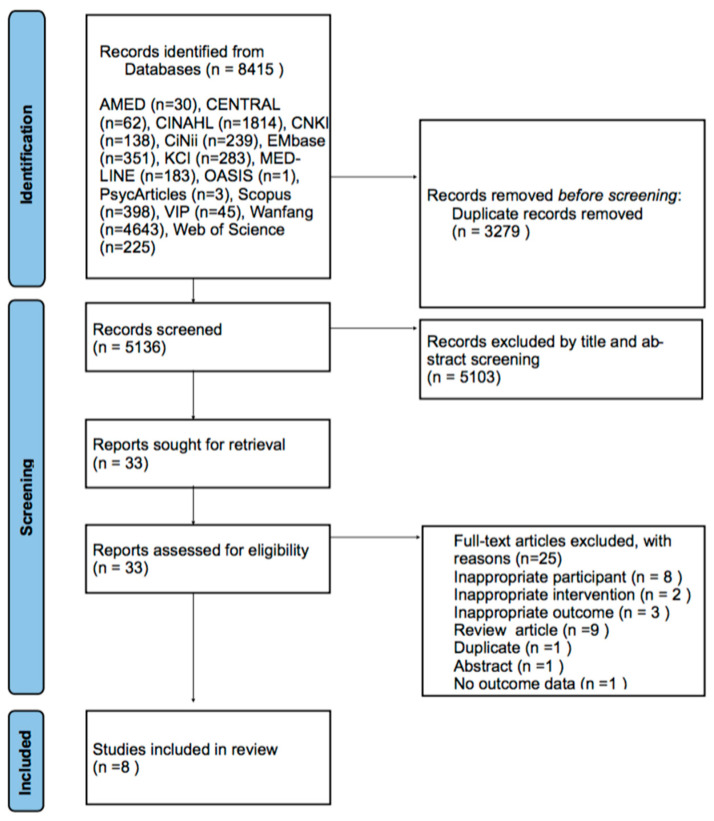
Flowchart of the identification and screening for the eligible studies.

**Table 1 healthcare-11-02957-t001:** General Characteristics of the included studies.

Study (Country)	Design	Sample Size (Included →Analyzed)	Mean Age (Range) (Years)	Sex (Male/Female)	Population (Diagnostic Tool)	Affiliation and Rank
King 2015 (USA) [32]	RCT	29 (15:14)→20 (12:8)	33.3 ± 6.1/32.8 ± 9.2	29:0	OEF/OIF veterans with PTSD (DSM-IV)	Active-duty Enlisted 18 Officer 2
Prisco 2013 (USA) [33]	RCT	35 (12:12:11)→25 (8:8:9)	37.8 ±11.4/37.9 ± 10.3/37.6 ± 8.0	25:10	OEF/OIF Veterans suffered from PTSD-related insomnia(DSM-IV-TR)	Enlisted 29 Officer 4 Warrant officer 2
Engel 2014 (USA) [34]	RCT	55 (28:27)→55 (28:27)	37.2 ± 11.3/32.6 ± 8.3	38:17	PTSD (DSM-IV)	Active duty Enlisted 39
Huang 2019 (USA) [35]	RCT	60 (30:30)→55 (27:28)→52 (25:27)	40.7 ± 10.8/40.3 ± 10.1	46:14	mTBI, PTSD (subgroup analysis)	Veterans receiving care from the Atlanta VAMC(Atlanta Veterans Affairs Medical Center) TBI clinic
Eisenloh 2010 (Germany) [36]	Before-after study	27→25	32.96 ± 9.66	21:6	History of trauma-related experiences (ICD-10)	Service women and men of the Bundeswehr Non-commissioned officers 15 Officer 6 Enlisted 6
Cronin 2013 (USA) [37]	Before-after study	5→5	29–48	3:2	Participants in the Veterans Sustainable Agriculture Training Program (VSAT). Four had combat experience and one had military sexual assault trauma. PTSD diagnosis was made in 4 people except 1 veteran	Army medic 1 Navy corpsman 1 Did not disclose their former military jobs 2 Who had experienced military sexual trauma 1
Arhin 2016 (USA) [38]	Case series	3→3	41–59	3:0	Patient of PTSD with tinnitus	NR
King 2016 (USA) [39]	Qualitative study	17→17	18–49	17:0	PTSD (DSM-IV)	Active-duty military personnel Marine 8 Navy 6 Army 3

Abbreviations: RC,: randomized controlled trial; DSM, Diagnostic and Statistical Manual of Mental Disorders; TR, text revision; PTSD, posttraumatic stress disorder; OEF/OIF, Operation Iraqi Freedom and Operation Enduring Freedom; mTBI, mild Trauma Brain Injury; ICD-10, International Classification of Diseases, Tenth Revision, NR, Not reported.

**Table 2 healthcare-11-02957-t002:** Characteristics of the included randomized controlled trials.

Study (Country)	Sample Size (Included →Analyzed)	(A) Treatment Intervention	(B) Control Intervention	Duration of Treatment/Follow Up	Outcome	Results Reported	Adverse Events Reported
King, 2015 (USA) [32]	29 (15:14)→20 (12:8)	Auricular acupuncture	Waiting list control	3 weeks	(1-1) Actigraphy- SOL (1-2) WASO (1-3) Sleep Efficiency (%) (1-4) NOA (1-5) Total Sleep Time (2-1) Consensus Sleep Diary-SOL (2-2) WASO (2-3) Sleep Efficiency(%) (2-4) NOA (2-5) Total Sleep Time (3) PCL-M (4) PHQ-9 (5-1)PSQI- Sleep quality (5-2) sleep latency (5-3) sleep duration (5-4) sleep efficiency (5-5) sleep disturbance (5-6) sleep medication (5-7) daytime dysfunction	(1-1) N.S (1-2) N.S (1-3) N.S (1-4) N.S (1-5) N.S (2-1) N.S (2-2) N.S (2-3) N.S (2-4) N.S (2-5) N.S (3) N.S (4) N.S (5-1) A) > (B) ++ (5-2) N.S (5-3) N.S (5-4) N.S (5-5) N.S (5-6) N.S (5-7) (A) > (B) ++	(A): 1 fall down. (B): 2 alcohol-related events, 1 wrist injury, 1 incident of suicidal ideation.
Prisco, 2013 (USA) [33]	35 (12:12:11)→25 (8:8:9)	(a) Auricular acupuncture	(b) Sham auricular acupuncture (c) WTL	2 months	(1-1) MSD—refreshness rating (1-2) soundness rating (2) ISI -Total Score (3-1) MSD—TST (3-2) SL (3-3) SE (3-4) Naps (4-1) Actigraphy—TST (4-2) SL (4-3) SE (4-4) Naps	(1-1) N.S (1-2) N.S (2) (a) > (b),(c) + after 1 month, N.S after 2 month (3-1) N.S (3-2) N.S (3-3) N.S (3-4) N.S (4-1) N.S (4-2) N.S (4-3) N.S (4-4) N.S	(a): dropped out due to discomfort with acupuncture (b),(c): none
Engel, 2014 (USA) [34]	55 (28:27)→55 (28:27)	Manual acupuncture + Usual PTSD care	Usual PTSD Care	4 weeks	(1) PCL (2) CAPS (3) NRS (4) BDI (5-1) SF-36 PCS (5-2) MCS	(1) (A) > (B) ++ (2) (A) > (B) + (3) (A) > (B) ++ (4) (A) > (B) ++ (5-1) (A) > (B) ++ (5-2) (A) > (B) ++	(A), (B): None
Huang, 2019 (USA) [35]	60 (30:30)→55 (27:28)→52 (25:27)	Manual acupuncture	Sham meridian acupuncture	1 month	(1) PSQI global score (2) Actigraphy sleep efficiency	(1) (A) > (B) ++ (2) (A) > (B) + PTSD status did not interact with this effect.	NR

“+” and “++” mean significant differences between two groups, *p*  <  0.05 and *p*  <  0.01, respectively. “N.S” means no significant difference between two groups, *p*  >  0.05. Abbreviations: BDI, total score on the Beck Depression Inventory-II; CAPS, Clinician-administered PTSD Scale total severity score; CI, confidence interval (95%); ISI, Insomnia Severity Index; mTBI, mild Trauma Brain Injury; MSD, Morin Sleep Diary; NOA, number of awakenings; NRS, average pain in the past week on the Numeric Rating Scale for Pain; PCL, Posttraumatic Stress Disorder Checklist score; PCL-M, Posttraumatic Stress Disorder Checklist—Military version; PHQ-9, Patient Health Questionnaire-9; PSQI, Pittsburgh Sleep Quality Index; PTSD, posttraumatic stress disorder; SE, Sleep Efficiency; SF36 MCS, Mental Health Component Summary score; SF36 PCS, Physical Component Summary score; SL, Sleep Latency; SOL, sleep onset latency; TST, Total Sleep Time; WA, Wrist Actigraph; WASO, wake after sleep onset.

**Table 3 healthcare-11-02957-t003:** Characteristics of the included before-after studies.

Study (Country)	Sample Size (Included →Analyzed)	Treatment Intervention	Duration of Treatment/Follow Up	Outcome	Results Reported	Adverse Events Reported
Eisenlohr 2010 [36]	27→25	Manual acupuncture + Auricular acupuncture	NR	(1) Subjective trauma-related symptoms (1-1) Sleep disorder (1-2) Restlessness (1-3) Agitation (1-4) Frightfulness (1-5) Aggression	(1-1) post treatment: improved * (1-2) post treatment: improved * (1-3) post treatment: improved * (1-4) post treatment: improved * (1-5) post treatment: improved *	Excessive sleep, drunken feeling; All improved after excluding specific acupoint.
Cronin 2013 (USA) [37]	5→5	Manual acupuncture	5 days/2 weeks	(1) PSQI (2) PCL-M	(1) post treatment: improved ^+^ follow up: improved ^+^ (2) post treatment: improved ^+^ follow up: improved ^+^	NR

“+” mean significant differences between two groups, *p*  <  0.05. “*” mean no estimated *p*-value for before-and-after comparison. Abbreviations: PCL-M, Posttraumatic Stress Disorder Checklist—Military Version; PSQI, Pittsburgh Sleep Quality Index; PTSD, posttraumatic stress disorder; NR, Not Reported.

**Table 4 healthcare-11-02957-t004:** Characteristics of the included case series study.

Study (Country)	Sample Size (Included →Analyzed)	Treatment Intervention	Duration of Treatment/Follow Up	Outcome	Results Reported
Arhin 2016 (USA) [38]	3→3	Manual acupuncture	Various	Subjective statement of the degree of improvement in tinnitus	In all cases, tinnitus improved after acupuncture, and preference for the treatment procedure had been identified. Associated PTSD symptoms also improved.

Abbreviations: PTSD, posttraumatic stress disorder.

**Table 5 healthcare-11-02957-t005:** Characteristics of the included qualitative study.

Study (Country)	Sample Size (Included →Analyzed)	Treatment Intervention	Duration of Treatment/Follow Up	Extracted Key Themes	Major Specific Statements
King 2016 (USA) [39]	17→17	Auricular acupuncture	3 weeks	(1) improved sleep (2) increased relaxation (3) decreased pain (4) loved/liked the auricular acupuncture treatments	(1) Ability to fall asleep faster, stay asleep longer, fall asleep during auricular acupuncture treatments, and experience fewer nightmares, improvement of daytime functioning (2) Increased relaxation both during and after receiving the auricular acupuncture treatments. (3) Improvements in low back pain, hip pain, neck pain, and headache (4) participant “loved” and “liked” the treatment and no negative written comments were found

**Table 6 healthcare-11-02957-t006:** Acupuncture intervention details according to the Standards for Reporting Interventions in Clinical Trials of Acupuncture guideline.

Study ID	Acupuncture Rationale	Practitioner Background	Details of Needling						Treatment Regimen	
			Type of Acupuncture	Number of Needle Insertions Per Participant Per Session	Location Of Points Used	Depth of Insertion	Response Sought	Needle Type	Number of Treatment Sessions	Frequency and Duration of Treatment Sessions
								Diameter, Length, and Manufacturer or Material		
King, 2015 [32]	Acupoint selection was based on review of previous AA insomnia studies and expert opinion.	Privileged military acupuncture provider, who had 2 years of clinical experience (over 500 treatments).	Auricular Acupuncture	24	Shenmen, Zero point, Brain, Thalamus Point, Pineal Gland, Master Cerebral, Insomnia 1, 2; Kidney C, Heart C, Occiput, Forehead.	NR	NR	0.20 mm diameter, 15 mm in length, D type needles, SEIRIN Corporation, Shizuoka, Japan)	9	30 min 3 times/week
Cronin 2013 [37]	NADA protocol	NR	Auricular Acupuncture	12	Shenmen, Sympathetic, Kidney, Liver and Lung/Heart	NR	NR	Seirin D-Type needles (40 gauge—red)	5	45 min 1 time/day
Prisco 2013 [32,39]	Using a Traditional Chinese Medicine (TCM) map as a guide, Selection of the points was based on acupuncture interventions for insomnia and the study acupuncturist’s experience with OEF/OIF veterans with PTSD-related insomnia.	All acupuncture services were performed in accordance with the established principles and practices of the National Certification Commission for Acupuncture and Oriental Medicine.	Auricular Acupuncture	10	Shenmen, Sympathetic, Kidney, Liver, Hippocampus	Needles inserted straight, reached the ear cartilage and to a depth where the needle could stand by itself. No guide tube used.	no needle manipulation.	DBC Brand Spring Handle Needles, size 0.16 × 15 mm	16	2 times/week 2 months
King 2016 [38]	Same as King, et al. [27]	Military nurse with supple- mental privileges to perform auricular acupuncture and had 2 years of clinical experience performing auricular acupuncture treatments.	Auricular Acupuncture	24	Shenmen, Zero point, Brain, Thalamus Point, Pineal Gland, Master Cerebral, Insomnia 1, 2; Kidney C, Heart C, Occiput, Forehead.	NR	NR	0.20 mm diameter, 15 mm in length, D type needles, SEIRIN Corporation, Shizuoka, Japan)	9	30 min 3 times/week
Eisenlohr 2010 [35]	NR	Trained acupuncturist with B diploma and 10 years of experience	Manual Acupuncture	NR	GV20, HT5, 7; ST23, SP6; LR3, PC6, 7; BL62, Extra point Anmian 1, 2; Extra point KH1; PT1, 2.	NR	NR	Cloud&Dragon 0.30 × 30 mm + 0.20 × 15 mm; Permanent needles Sedatelec ASP steel with stimulation magnet	NR	35 min 2–3 times/week
Arhin 2016 [37]	Korean four-needle technique	NR	Manual Acupuncture	4	NR	NR	NR	NR	4–6	1 time/2 weeks
Engel 2014 [33]	First 4 sessions were standardized for all participants, and the last 4 sessions allowed acupuncturists flexibility to individualize based on standard diagnostic criteria (pulse, tongue, symptoms, color, odor, etc).	Acupuncturists were licensed, practicing regularly, and recipients of advanced degrees (M.Ac.) from the Tai Sophia Institute for the Healing Arts, a program emphasizing traditional Chinese medicine philosophies and using 5 elements theory as overall guide to treatment.	Manual Acupuncture + Auricular Acupuncture	NR	BL 13, 14, 15, 18, 20, 23; LR 3; LI4; HT5, 7; PC 6; KI3, 9; Ren 4, 15; Du 24; Ear Shenmen; Yintang. (Used at least 1 time)	NR	NR	Seirin brand, J type: 0.14, 0.16, and 0.2 mm and L type: 0.2 mm	8	15–30 min (depending on point prescription) 2 times/week
Huang 2019 [34]	Most commonly used acupoints in prior sleep studies for treatments	Board-certified physiatrist with advanced training in acupuncture	Meridian Acupuncture + Auricular Acupuncture	Various	Auricular shenmen, PC-6, SP-6; Other points are selected according to standardized principles according to the symptoms of the therapist.	NR	no needle manipulation.	Sterile, 34-gauge single-use disposable metal real or sham acupuncture needles	6–10	30 min 2 times/week

Abbreviations: AA, Auricular Acupuncture; NADA, National Acupuncture Detoxification Association; NR, Not reported; OEF/OIF, Operation Iraqi Freedom and Operation Enduring Freedom.

## Data Availability

On reasonable request, the authors will provide access to the data.

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
