# Peer review of "Acupuncture Therapy for Military Veterans Suffering from Posttraumatic Stress Disorder and Related Symptoms: A Scoping Review of Clinical Studies"

_healthcare, 2023, doi:10.3390/healthcare11222957_

Round 1
Reviewer 1 Report
Comments and Suggestions for Authors
The variety of studies (RCT, before/after and case series) make conclusions on this evidence hard to synthesize. However, asking the series of questions and reviewing the literature that way was valuable to get a summary of the body of literature for the use of acupuncture for PTSD.
1. Typos are found in the abstract. Please edit
2. Unclear when the term Veterans are used, is the population referring to US Veterans or Korean Veterans or both? I think it may be of use to describe who the Veteran population is in this case. Looks like references to prevalence of PTSD reflects the US Veteran population due to the description of the war/conflict name.
3. Battlefield Acupuncture is two words not three.
4. It may be useful to use the terms of active duty military and veterans in describing the study population. Active duty are considered a different population that veterans and the results may be slightly different between the two group as motivations/incentives for recovery may be different and the age of the participants is typically different. One of the RCTs was on an active duty population.
5. Were age demographics studied?
6. The editing might benefit from someone who can make the distinction of active duty populations (sometimes called soldiers-line 496) and veterans. The language should be cleaned up to accurately call this population active military and veterans. Soldiers refers to Army service members but not the other branches. Military medicine refers to the active duty population not veterans.
Comments on the Quality of English LanguageThere is a lot of confusion in terminology of military populations and veteran populations that should be cleaned up to be clear and accurate. Some service members- and veterans may be bothered by the lack of understanding of the use of terminology for this manuscript.
Author Response
I attached file.

Reviewer 2 Report
Comments and Suggestions for Authors
esteemed author
Thank you for your work. It has been a pleasure to read.
I would like to make a few comments with the intention of improving the quality of your work.
You present an appropriate design; the results are appropriate to your design and the implications for practice are very positive.
However, if we look at the results of your study, of the more than 5000 articles initially selected, there are only 8 which, when analysed, are less than 10 years old. This points me to defects in the search, in particular in the criteria for selecting the sample; would it be possible to modify the criteria for including the papers, and include a time criterion, which restricts the number of articles initially included? if not, could you duly justify the reason for selecting all the papers published in the databases from the time of their creation?
I hope you can resolve these details.
Thank you
Author Response
I attached file.

Reviewer 3 Report
Comments and Suggestions for Authors
This Reviewer commends the Authors for a well-written analyses in the very important and emerging domain of an alternative medicine/therapy: acupuncture. The study findings and recommendations can provide great ‘best practices’ for this emergent area and the Authors performed a very credible, disciplined and thorough analysis. The following recommendations are for the Authors to consider making to their existing manuscript:
Page 1, Line 31 Abstract: Abstract reads well and conveys to the Reader the relevance and importance of this topic. Recommend re-writing the following sentence on Line 31 from: “The treatment period varied between 5 day and 2months.” to “The treatment period varied between 5 days and 2 months.” By adding the “s” at the end of “day” and inserting a space between “2” and “months” this corrects the errors.
Page 2, Line 54: Recommend adding an “s” at the end of “decrease” for accuracy. The new word should read as follows: “decreases”.
Page 2, Lines 91-94: The authors state that “A scoping review is a methodology used to identify certain characteristics or concepts of studies and conduct mapping and discussion about these characteristics or concepts, rather than presenting answers to specific questions [26].” Recommend the Authors add one or two more sentences here describing the difference between a scoping review, and a systematic review as the Authors recommend a future systematic review in Line 99.
Page 24, Line 547: Recommend removing the underline symbol between the words “studies” and “The”. The new sentence fragment should read as follows: “studies. The . . .”
Page 24, Line 554: Recommend removing the underline symbol at the end of the sentence. The new sentence fragment should read as follows: “ . . . in previous studies.”
This Reviewer looks forward not only to the publication of this study, but additionally future follow-on studies on this topic by these talented Authors.
Thank you.
Author Response
I attached file.
